# To Rip or not to Rip: A Reinforcement Learning-based Rip-up and Reroute Algorithm for Global Routing

**Fangzhou WANG**
Department of Computer Science and Engineering
The Chinese University of Hong Kong
Shatin, NT, Hong Kong
fzwang@cse.cuhk.edu.hk

**Wanying ZHENG**
Department of Computer Science and Engineering
The Chinese University of Hong Kong
Shatin, NT, Hong Kong
wyzheng@cse.cuhk.edu.hk

## Abstract

Routing, including global routing and detailed routing, has been a critical step in the design of integrated circuits. Most of the existing global routers will firstly use techniques like pattern routing and layer assignment to quickly generate a routing solution and optimize total wirelength and via usage. Then rip-up and reroute (RRR) scheme will be applied to iteratively reduce the number of overflows in the whole design. However, compared with initial routing stage, RRR will be much more time consuming. It will rip up all the nets that pass through overflowed area and reroute them sequentially. Even if the routing resources in one routing cell is overused by 1, the router will rip up all the nets that are routed on the routing cell, as it does not know which net will be the best choice to rip up. In this way, RRR may be doing a lot of redundant work. Besides, some initial routing solutions that are optimal in terms of wirelength will also be wasted when they are ripped up, causing a loss of routing quality. Therefore, in this project, we propose to use reinforcement learning to help decide which nets to rip up in each RRR iterations. An actor-critic based Proximal Policy Optimization (PPO) agent is trained for this task. Experimental results show that the proposed approach can successfully reduce the number of rerouted nets with little loss of routing quality on the ICCAD'19 global routing contest benchmarks, which demonstrate the effectiveness of our model. The presentation link can be found here[1].

## 1 Introduction

Routing has been an important problem in the field of very large scale integration (VLSI) physical design for several decades. After floorplanning and placement, routing will decide the path to interconnect pins of the same signal net. Due to the complexity of routing problem, it is usually decomposed into two sub-problems, namely global routing and detailed routing. By global routing, a coarse-grained routing plan will be generated, minimizing total wire length, number of overflows, or other objectives. After that, based on the result of global routing, detailed routing will be conducted to assign routes to specific routing tracks while honoring various complicated design rules.

---

[1] https://drive.google.com/file/d/1b94l5C6tZHlhvVleAMannV7LdyzYloxQ/view?usp=sharing

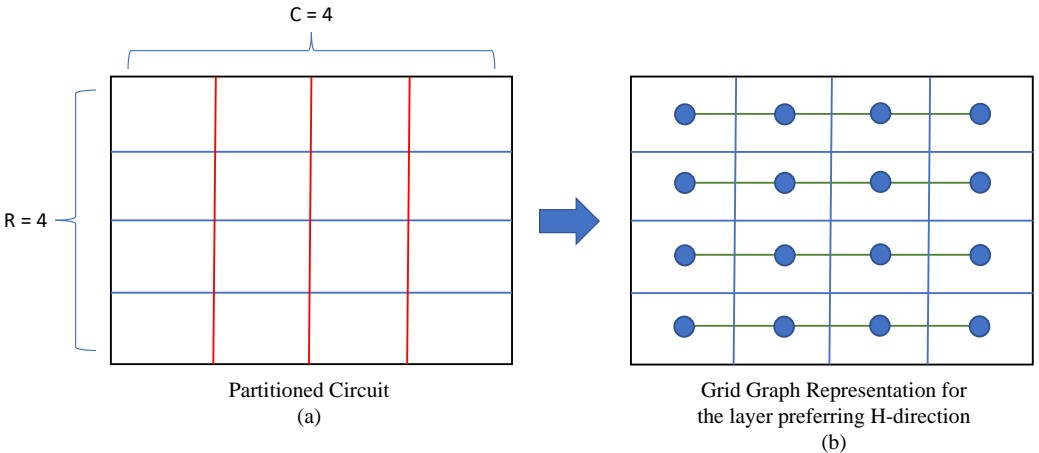

Figure 1: Gcell to grid graph conversion.

In the global routing problem, the whole circuit is divided into many small rectangular regions by predefined grid lines that are evenly distributed, like Figure 1(a) shows. As the routing is done in a 3D routing spacing, we can assume there are $R \times C \times L$ regions, where $L$ refers to the number of routing layers. Those small regions are called global routing cells (Gcells). It is worth noticing that each layer has its own preferred routing direction, which can be either horizontal or vertical (H or V). In-layer routing must follow the preferred routing direction. By considering each Gcell as a vertex and creating edges between adjacent Gcells, we can obtain a 3D grid graph. Here we consider two Gcells adjacent to each other if (1) they have the same $(row, col)$ number and they are in adjacent layer or (2) they are on the same layer and adjacent to each other in the preferred direction. We call edges created under condition (1) *via edges* and those created under condition (2) *wire edge*. The capacity of a wire edge equals to the number of nets that can go through the edge while the capacity of a via edge is normally handled in different ways. Figure 1(b) gives an example of the edge modeling on a layer whose preferred routing direction is horizontal.

The global routing problem is usually defined as, given a grid graph $G(V, E)$ and a set of nets $N$ to be connected, find a path for each net such that the total wirelength is minimized and the number of overflows is minimized. Here the overflow on an edge $e$ is calculated as $max(0, demand(e) - supply(e))$, which measures the overuse of routing resource. Figure 2 shows an example routing result for one net on a grid graph, where pins are connected by edges marked in red.

Then comes to the definition of rip-up and reroute (RRR) problem in this global routing context. Originally in the initial routing, routing paths are normally generated to minimize the total wirelength and the number of vias. Therefore, overflows will inevitably appear after the initial routing stage. The term "rip up" refers to the action that we discard the routing solution of certain net and reclaim the routing resources that have been used by the net. In this way, we can first make those overflows disappear by ripping up some nets and later replan the routing solution for those rip-upped nets. Formally, after all the nets in $N$ have been routed by initial routing, we will start the RRR stage, which may consist of several iterations, each of which can be defined as the following:

1. Check each edge in $E$ and define all the nets that pass through overflowed edges as $N_{vio}$.

2. Choose the nets to be ripped up and define them as $N_{rip}$.

3. Decide the net ordering for nets in $N_{rip}$ and reroute them with different reroute schemes, which can be either in one-by-one manner or in rip-up-all manner.

In the one-by-one manner, nets will be ripped up and rerouted one by one. Meanwhile, in the rip-up-all manner, all the nets in $N_{rip}$ will be ripped up first and then rerouted sequentially. During RRR, nets will normally be rerouted using maze routing, which tends to produce routing solutions with less overflow but larger wirelength.

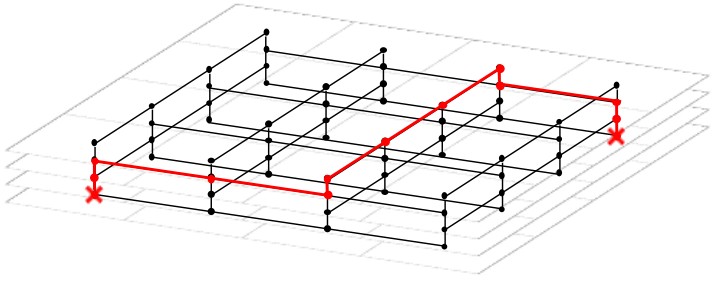

✖ Pin      ⋮ Via Edge      •——• Wire Edge

Figure 2: An example routing solution for one net on a grid graph.

In almost all the implementations of RRR, the $N_{rip}$ in step two equals $N_{vio}$. In this way, RRR will iteratively reduce the total number of overflows while increasing wirelength. The final goal of RRR is to minimize a metric considering both the total routed wirelength and the total number of overflows. The metric $Score$ can be described by,

$$Score = \alpha \cdot WL_{wire} + \beta \cdot WL_{via} + \gamma \cdot shortArea, \tag{1a}$$

$$WL_{wire} = \sum_{n \in N} wirelength(n), \tag{1b}$$

$$WL_{via} = \sum_{n \in N} num_{via}(n), \tag{1c}$$

where $Score$ is a weighted sum of $WL_{wire}, WL_{via}$ and $shortArea$. Weights are represented by $\alpha, \beta$, and $\gamma$ respectively. In the above, $shortArea$ refers to the expected short area in the routing solution, which is calculated by CUGR [2] and is positively correlated with the number of overflows. $wirelength(n)$ and $num_{via}(n)$ denote the routed wirelength and the number of via edge used by net $n$ separately.

It is observed that the traditional RRR flow may involve a lot of unnecessary computation, which will be discussed in Section 2. Therefore, in this project, we explore a reinforcement learning-based RRR scheme, which is in the one-by-one manner, aiming to reduce the total number of rerouted nets without causing much routing quality loss.

## 2   Related Works

Various RRR techniques exist as an important component in most of the existing global routers[1, 2, 3, 7], most of which are inspired by the negotiation-based RRR scheme introduced in PathFinder[4]. For instance, NTHU-Route 2.0[1] adopts a new net ordering scheme for RRR after finding all the nets that pass through overflowed area while NCTU-GR 2.0[3] adopts a collision-aware RRR scheme to improve the effectiveness of multi-threaded maze routing. However, those works either focus on the net ordering or cost scheme for maze routing. Another potential to improve the RRR scheme may lie in the strategy for deciding whether to rip up a net or not. Based on the fact that the wirelength produced by initial routing is always shorter than that produced by maze routing, ripping up all the nets in $N_{vio}$ may lead to considerable increase in total wirelength than simply rip up part of the nets in $N_{vio}$.

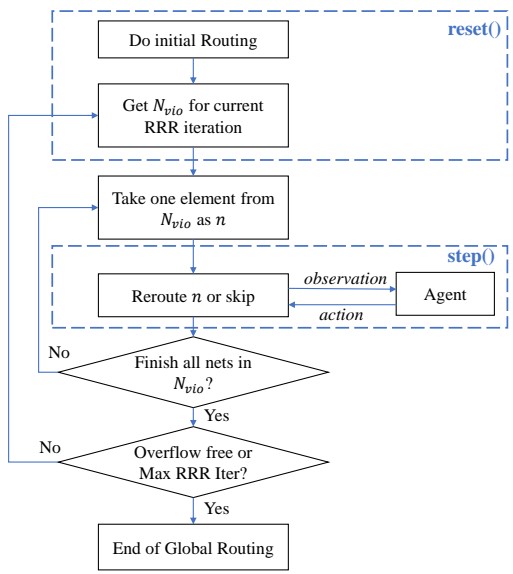

Figure 3: The environment overview. Note that "Iter" represents "Iteration".

## 3  Environment Setup

To enable the interaction between our reinforcement learning and a global router, we build an environment called Global Routing Environemnt (GRE), where necessary APIs like reset(), step() are defined. Adapted from CUGR [2], which is the state-of-the-art academic global router, our environment can finish a complete flow of global routing and report the final score to evaluate the performance of the trained agents. Figure 3 illustrates the overview of GRE.

### 3.1  Environment Initialization

Like a typical environment for reinforcement learning, GRE will be initialized before the agent can get observations and take actions. The initialization corresponds to the initial routing stage in CUGR. When "reset()" is called, GRE will parse the corresponding benchmark and perform the built-in initial routing algorithm to construct a routing solution for all nets. After that, the environment will be in a state after intial routing and before the first round of RRR.

### 3.2  Agent

The task of an agent is to decide whether to rip up and reroute current net $n_{cur}$ or not, based on the observations related to $n_{cur}$. It is worth noting that both RRR iteration number and $n_{cur}$ will be maintained in GRE such that each net in current $N_{vio}$ can be processed by the agent.

### 3.3  State and Observation

The state of the environment can quite complicated, which consists of (1) current routing solution for all nets, (2) the supply and demand for all edges, (3) $N_{vio}$ in current RRR iteration and (4) current net to be considered $n$. Thus, it might not practical to directly show the state to our agent, for many unrelated information in the usage of edges and complex graph topology of the routing solution. Instead, from the current state, we will extract representative information and use them as the observation.

In each step, we will consider one net only. Thus the observation will correspond to $n$ and can be described by,

$$\mathcal{O} = <wirelength(n), D_{ovfl}(n), D_{comp}(n)>, \tag{2a}$$

$$D_{ovfl}(n) = \sum_{e \in E_n} EstOverflow(e), \tag{2b}$$

$$D_{comp}(n) = \frac{\sum_{n' \in N} Area(BBox(n) \cap BBox(n'))}{Area(BBox(n))}, \tag{2c}$$

where $\mathcal{O}$ denotes the observation (feature vector) for net $n$. In Equation (2), $D_{ovfl}(n)$, namely the overflow degree, measures the total number of estimated overflows associated with net $n$ in current routing solution. If a net has large overflow degree, it may pass through a lot of highly congested edges and has also made its contribution to the congestion. Meanwhile, $D_{comp}(n)$, namely the competition degree, measures how intense the routing resource competition can be between net $n$ and the rest of the nets. $BBox(n)$ denotes the 2D bounding box of net $n$, which is determined by net's pin locations. A larger competition degree may indicate that the net will have less abundant resources to use. It is worth noting that the observation $\mathcal{O}$ only contains information about net $n$ in its local region, which is designed in this way as nets that are far away will not influence each other.

### 3.4 Action

After initialization, GRE will internally find the set of nets that pass through overflowed edges and store them as $N_{vio}$. Nets in $N_{vio}$ will be processed sequentially. At the beginning of each RRR iteration, $n$ will be assigned as the first net in $N_{vio}$. After observations related to net $n$ are fed into the agent, the action will be to rip up and reroute net $n$ or skip it, which corresponds to the "step()" part in Figure 3.

### 3.5 Reward

We use $score_t$, $vio_t$ to denote the $Score$ which is defined in Equation (1a) and the summation of $D_{ovfl}(n)$ for all nets after step $t$. We then define $score_0$, $vio_0$ to be the state right after environment initialization, that is, after initial routing and before the first round of RRR. Let $a_t$ be the action at step $t$. The reward after step $t$ will be denoted as $r_t$,

$$r_t = (score_{t-1} - score_t) + 100 \times (vio_{t-1} - vio_t) + 10 \times \mathbb{1}(a_t == skip), \tag{3a}$$

where $\mathbb{1} \in \{0, 1\}$ is the indicator function.

## 4 Proposed Approach

We propose to use policy gradient methods, since the large number of nets to be chosen for rip-up results in high dimensional action spaces, it is tedious to learn the action-value function while using action-value methods. A parameterized policy can be learnt based on the observation on congestion map, current routing solution for certain net and the routing solution for the rest of nets, or some features extracted from the current routing solution, like some vectors of scores calculated for the solution using heuristics. To improve training stability, we propose to use proximal policy optimization (PPO)[6]. The reward may take the number of nets to rip-up, overflows, via edges, and total wirelength into account.To reduce action space, we may decide the net ordering first by pin numbers and process those nets one-by-one following this order.

We summarize the steps to train our actor-critic network, which is shown in Algorithm 1.

The goal of our actor-critic network is to maximize the discounted overall rewards in an episode which consists of a number of RRR iterations. A full episode is finished when either reaching maximum number of RRR iterations or having rerouted all .The objective function can be formulated as follows:

$$\mathbf{J}(\theta) = \mathbf{E}_{\tau \sim \pi_\theta}[R(\tau)] \tag{4}$$

**Algorithm 1** Training steps of our model

---

1: Input: initial policy parameters $\theta_0$, initial value function parameters $\phi_0$
2: **for** $k = 1, 2, ...$ **do**
3:     Collect set of trajectories $\mathcal{D}_k = \{\tau_i\}$ by running policy $\pi_k = \pi(\theta_k)$ in the environment
4:     Compute returns $\hat{\mathcal{R}}_t$.
5:     Compute advantage estimates $\hat{A}_t$ based on the current value function $V_{\phi_k}$
6:     Update the policy by maximizing the PPO-Clip objective:

$$\theta_{k+1} = \underset{\theta}{\arg\max} \frac{1}{|\mathcal{D}_k|T} \sum_{t \in \mathcal{D}_k} \sum_{t=0}^{T} \min \left( \frac{\pi_\theta(a_t|s_t)}{\pi_{\theta_k}(a_t|s_t)} A^{\pi_{\theta_k}}(s_t, a_t), g(\epsilon, A^{\pi_{\theta_k}}(s_t, a_t)) \right)$$

    Fit value function by regression on mean absolute error:

$$\phi_{k+1} = \underset{\phi}{\arg\max} \frac{1}{|\mathcal{D}_k|T} \sum_{\tau \in \mathcal{D}_k} \sum_{t=0}^{T} |V_\phi(s_t) - \hat{\mathcal{R}}_t|,$$

    via gradient descent with Adam

---

where $\tau$ indicates an episode, $\pi_\theta$ is the policy parameterized by $\theta$, $R(\tau)$ is the return function on $\tau$, which is the sum of discounted rewards the episode, defined as follows:

$$R(\tau) = \sum_{i=1}^{T} \gamma^{i-k} r_t \tag{5}$$

where $\gamma$ is discount factor.

In our model, the actor and network are both constructed by a three-layer MLP model with similar settings, only different in output space. The actor network outputs the action to take, and the log probability of the selected action in the distribution. For the critic network, it only outputs a scalar. We normalize the observation using running estimates of mean and standard deviation, as the observation of wire length is very large in scale, which may probably dominate the input for the actor. The running mean and standard deviation is also saved for each case as part of the actor model, as the running statistics will vary according to the scale of different benchmarks. The benchmarks are of different sizes, different number of nets, layers, which will result in different scale of wire lengths, via numbers, etc.

The reward is the difference of the weighted sum of the total wire lengths, via numbers, short areas before and after the action. As the difference is the quantified change in the overall routing plan, it can be viewed as the reward or penalty for this decision.

Advantage function is defined as:

$$A^\pi(s, a) = Q^\pi(s, a) - V_{\phi_k}(s) \tag{6}$$

where $Q^\pi$ is the Q-value of state action pair (s,a), and $V_\phi$ is the value for some observation $s$ given by our critic network based on the parameters $\phi$ on the $k$-th iteration.

We combine the actor and critic losses and an entropy bonus to the objective loss function to encourage more exploration, controlled by an entropy coefficient parameter. Our objective loss function is defined as follows:

$$L_t^{CLIP+V+S}(Q) = \hat{E}_t \left[ L_t^{CLIP}(Q) + c_1 L^V(Q) - c_2 S[\pi_Q](s_t) \right] \tag{7}$$

where $c_1$ and $c_2$ are coefficients, $S$ denotes an entropy bonus.

## 5  Experimental Results

The global routing environment is adapted from CUGR [2], which is mainly written in C++. To allow smooth interaction between the global routing environment and the reinforcement learning agent, we use pybind11[2] to wrap the modified C++ code so that GRE can be instantiated as a Python object and we do not need to rewrite the whole global routing logic in python. Lastly, our reinforcement learning agent is implemented using PyTorch [5].

---

[2]https://pybind11.readthedocs.io

Table 1: Experiment Results on Benchmark 18test3

| RRR Algorithm | #(Rerouted nets) | $WL_{wire}$ | $WL_{via}$ | shortArea | *Score* |
|---|---|---|---|---|---|
| Rip-Up-All | 157 | 8538671 | 300623 | 25.94 | 5484796.7 |
| One-by-One | 157 | 8541170 | 300647 | 25.94 | 5486141.8 |
| RandomPolicy | **71** | **8536815** | 300379 | 19.95 | 5479898.5 |
| RL-based | 82 | 8538058 | **300363** | **15.96** | **5478462.5** |

* Benchmark 18test3 is of size $329 \times 247 \times 9$ ($R \times C \times L$) and contains 36700 nets.

Experiments were conducted on the ICCAD'19 global routing contest benchmarks, which contain both tiny cases and large cases. For tiny cases, the routing will end immediately after initial routing as the routing solution is already violation free as the design is small and not congested. For large cases, one complete run of global routing will take several minutes, which makes it hard for the agent to be trained efficiently. Therefore, we picked the benchmark 18test3, which is of medium size, for the training purpose. The benchmark contains around 37K nets with a layout size of $329 \times 247 \times 9$.

For comparison, we implemented the following baseline RRR algorithms:

- Rip-Up-All: We firstly rip up all the nets in $N_{vio}$ and reroute nets sequentially, which is the original RRR algorithm in CUGR.
- One-by-One: Each net in $N_{vio}$ will be ripped up and rerouted one by one.
- RandomPolicy: Each net in $N_{vio}$ will be processed in one-by-one manner, with each net having a $\frac{1}{2}$ chance of being ripped up and rerouted.

Quantitative results are listed in Table 1. Our proposed reinforcement learning-based RRR algorithm is denoted as RL-based in the table. Compared with the Rip-Up-All and the One-by-One approach, our RL-based algorithm can better total score with both smaller total wirelength and *shortArea*. Meanwhile, with the RL-based algorithm, the total number of rerouted nets is greatly reduced compared with the first two approaches.

## 6 Conclusion

In this project, we propose an reinforcement learning-based rip-up and reroute (RRR) algorithm for global routing. Compared with the original RRR scheme in the state-of-the-art academic global router, our algorithm can effectively reduce the total number of rerouted nets and achieve slightly better routing quality on a benchmark of medium size from the ICCAD'19 global routing contest.

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
