# OpenReview forum: "To Rip or not to Rip: A Reinforcement Learning-based Rip-up and Reroute Algorithm for Global Routing"
_CUHK.edu.hk/2021/Course/IERG5350_

### Official Review · AnonReviewer3 · 2020-12-18
**Interesting Work**

**Rating:** 9
**Confidence:** 4

**Review:**

Evaluation of the quality:

The main contribution of this work is proposing Reinforcement Learning based method for Global Routing. It is an excellent work as the explanation of traditional and proposed method of global routing is clear enough. It is difficult to present the practical problem from real life to virtual environment so that it is a good way to simplify the large scale circuit and finish the specific tasks.

Clarity:
1. The author has made some example routing solutions for the readers to have better understanding in routing.
2. The environment overview is clear to show the working flow of Global Routing Environment made by the author.

Originality:

It is an excellent work as the author design their own grid graph and divide the standard of experiment results into different part.

Significance:

This work proposed a Reinforcement Learning based method for Global Routing. It may helps to solve the routing problem in real life. As it can reduce the reduce the total number of rerouted nets and achieve slightly better routing quality on a benchmark of medium size from the ICCAD’19 global routing contest comparing with the original RRR scheme.

Pros
1. The explanation of the algorithm is clear
2. The readers can have a good understanding in routing problem even they are not familiar with such issue.

Cons:
1. In Table 1, the weight of different parts for calculating the score can be further elaborate

---

### Official Review · AnonReviewer2 · 2020-12-20
**Good adaptation of RL onto RRR**

**Rating:** 8
**Confidence:** 4

**Review:**

The paper describes a reinforcement learning framework proposed for the rip-up and reroute (RRR) scheme that aims to improve the routing performance by helping to decide on the nets to rip up in each RRR iterations.

Personally I have no background knowledge on this topic of circuits and routing, so I could only provide general comments on the work. The overall presentation is clear and figures are provided for illustration when necessary. The PPO algorithm is standard with adaptations made to the state representations and pin number ordering optimization to fit the algorithm into the RRR problem. The experimental results are also promising, showing that the RL model indeed improves the performance while maintaining a low number of rerouted nets.

---

### Official Review · AnonReviewer1 · 2020-12-20
**Good question formulation but lack of experiments**

**Rating:** 7
**Confidence:** 3

**Review:**

Summary:

The authors implement the PPO to solve the rip-up and reroute problem in global routing. Firstly the authors formulated the problem into the reinforcement learning problem with the design of state, action, and reward. Then they used PPO to train the model and get better performance rather than the traditional method.

Pros:

1. Innovative: It is a creative idea of using RL to solve this problem, according to the related work, it seems that the authors are the first one using the AI algorithm to solve this problem.
2. Good formulation: It is a good job that the authors translated the routing problem into the RL problem. Each element is well designed, especially the state and reward.
3. Good environment building: It seems hard to build the environment with a combination C++ and Python, but the authors did it well.

Cons:
1. Lack of detail about the experiment: there is less detail introduction about the experiment, such as the selection of activation function of the MLP, the distribution of actor, the hyperparameters of the training, etc.
2. Lack of comparison of the other RL algorithm: The authors did not explain more about why they use the PPO, instead of the other algorithm, like DDPG, TD3.
3. Not sensitive improvement: It seems that the RL model selection and MLP structure is casual, and the results showed that there only a bit of improvement rather than the traditional method. And this paper did not mention how to design the weights of the score,  is there a standard method to set them? Because without the setting introduction,  I am confused that if you reached the better performance than RandomPolicy by setting the weights.(In section 5, it seems that only the shortArea have great improvement)
4. Lack of learning curve in experimental results.
5. Not well organized: the authors introduce the RL part of the problem roughly, I think they should introduce more detail about the RL model.

Evaluation:

1. Do more experiments and compare the performance of different algorithms and introduce the detail of the experiments.
2. Add more detail about the model description, like the bound of observation space, the action space, the reason why select the PPO, etc.
3. Design efficient neural networks and select the right hyperparameters to get more improvement.